# ConViTac: Aligning Visual-Tactile Fusion with Contrastive Representations

Zhiyuan Wu, Yongqiang Zhao, and Shan Luo

*Abstract*— We propose ConViTac, a visual-tactile representation learning network designed to enhance the alignment of features during fusion using contrastive representations. Our key contribution is a Contrastive Embedding Conditioning (CEC) mechanism that leverages a contrastive encoder pretrained through self-supervised contrastive learning to project visual and tactile inputs into unified latent embeddings. These embeddings are used to couple visual-tactile feature fusion through cross-modal attention, aiming at aligning the unified representations and enhancing performance on downstream tasks. We conduct extensive experiments to demonstrate the superiority of ConViTac in real world over current state-of-the-art methods and the effectiveness of our proposed CEC mechanism, which improves accuracy by up to 12.0% in material classification and grasping prediction tasks.

## I. INTRODUCTION

Vision and touch are two fundamental sensory modalities for robots, offering complementary information that enhances perception and manipulation tasks. Previous research has attempted to jointly learn visual-tactile representations to extract more meaningful information. However, these approaches often rely on direct combination, such as feature addition and concatenation, for modality fusion, which tend to result in poor feature integration.

The human brain demonstrates a remarkable ability to integrate visual and tactile information by visually pinpointing the area being touched and using tactile perceptions to enhance its understanding of that specific region within the visual field, aided by pre-learned semantic knowledge [1]. In this paper, inspired by this idea in neuroscience, we introduce ConViTac, a novel visual-tactile representation learning method that improves multi-modal fusion through contrastive representations, due to their potentials to build connectivity between different modalities [2]. Central to our approach is the Contrastive Embedding Conditioning (CEC) mechanism, which integrates contrastive representations, originally obtained through self-supervised learning, into a **fully supervised** learning framework. CEC mechanism first train a contrastive encoder through SimCLR [3] in a self-supervised way, and then leverages this pretrained contrastive encoder to project visual and tactile data into a learned joint space to get unified representations. These projected contrastive embeddings are then combined and used as a condition to align feature fusion via cross-modal attention. Our experimental results demonstrate the superiority of ConViTac

Department of Engineering, King's College London, Strand, London, WC2R 2LS, United Kingdom, {zhiyuan.1.wu, yongqiang.zhao, shan.luo}@kcl.ac.uk.

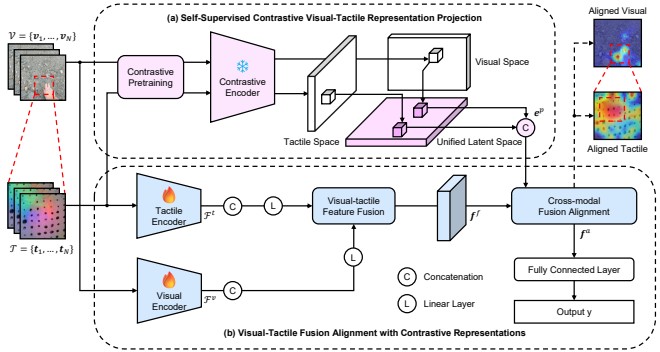

Fig. 1. An overview of our proposed ConViTac architecture. Our ConViTac processes visual and tactile sequences through a two-stage **C**ontrastive **E**mbedding **C**onditioning (CEC) mechanism: a) First, it performs self-supervised contrastive learning via SimCLR [3] on all visual-tactile data to pretrain a contrastive encoder, which projects both visual and tactile inputs into a unified latent space. b) Subsequently, it employs the projected contrastive representations to align visual-tactile fusion in supervised learning, with the contrastive encoder frozen. The output $y$ represents the results from various substream tasks, including material classification and grasping prediction. Aligned features are visualized by GradCam [4] for reference.

over existing state-of-the-art (SoTA) supervised and contrastive learning methods. We also perform ablation studies to quantitatively and qualitatively validate the effectiveness of the proposed CEC mechanism.

## II. SELF-SUPERVISED CONTRASTIVE MULTI-MODAL REPRESENTATION PROJECTION

We begin by training a contrastive encoder $\mathcal{E}^c$ within all visual-tactile data using self-supervised contrastive learning to project visual and tactile data into latent embeddings within a shared feature space, which is achieved through SimCLR [3]. During the self-supervised contrastive learning process, we employ $\mathcal{E}^c$ to obtain a projected latent embedding $e^p \in \mathbb{R}^{2N \times P \times D}$ from any single image $v$ in the visual sequence $\mathcal{V}$ and any single image $t$ in the tactile sequence $\mathcal{T}$:

$$e^p = C[\mathcal{E}^c(v), \mathcal{E}^c(t)]. \tag{1}$$

The loss function for the self-supervised learning $\mathcal{L}_c$ can be written as follows:

$$\mathcal{L}_c = -\sum_{i=1}^{2B} \log \frac{\exp(S_{i,i+B})}{\sum_{j \neq i} \exp(S_{i,j})}, \tag{2}$$

where $B$ represents the batch size of all samples, and $S_{i,j}$ refers to the similarity matrix, calculated as:

$$S_{i,j} = \frac{e_i^p \cdot e_j^p}{\tau}, \tag{3}$$

TABLE I

**MATERIAL CLASSIFICATION.** THE METRIC IS ACCURACY (%) AND THE BETTER RESULTS ARE IN BOLD FONT. CHANCE REFERS TO THE BASELINE
PERFORMANCE LEVEL THAT WOULD BE EXPECTED BY A RANDOM CLASSIFIER.

| Category | Methods | Modality | | *Touch and Go* | | | *ObjectFolder Real* | | |
| | | Vision | Touch | Category | Hard / Soft | Rough / Smooth | Category | Hard / Soft | Rough / Smooth |
|---|---|---|---|---|---|---|---|---|---|
| - | Chance | - | - | 18.6 | 66.1 | 56.3 | 13.8 | 50.6 | 49.0 |
| Contrastive Learning | VT CMC [5] | ✓ | ✓ | 68.6 | 87.1 | 82.4 | 29.5 | 59.8 | 65.6 |
| | SSVTP [6] | ✓ | ✓ | 70.7 | 88.6 | 83.6 | 34.1 | 61.1 | 74.3 |
| | MViTac [7] | ✓ | ✓ | 74.9 | 91.8 | 84.1 | 30.8 | 61.4 | 70.7 |
| Supervised Learning | STAM [8] | - | ✓ | 52.6 | 88.9 | 75.1 | 40.4 | 67.0 | 68.6 |
| | VTFSA [9] | ✓ | ✓ | 66.8 | 92.5 | 82.2 | 47.9 | 72.2 | 74.1 |
| | ConViTac (**Ours**) | ✓ | ✓ | **86.3** | **94.3** | **88.5** | **59.9** | **77.2** | **81.1** |

TABLE II

**PREDICTING SUCCESS OF GRASPING.** THE METRIC IS ACCURACY (%)
AND THE BEST RESULTS ARE IN BOLD FONT. CHANCE REFERS TO THE
BASELINE PERFORMANCE LEVEL THAT WOULD BE EXPECTED BY A
RANDOM CLASSIFIER.

| Category | Methods | Modality | | Accuracy (%) |
| | | Vision | Touch | |
|---|---|---|---|---|
| - | Chance | - | - | 50.8 |
| Contrastive Learning | VT CMC [5] | ✓ | ✓ | 56.3 |
| | SSVTP [6] | ✓ | ✓ | 59.9 |
| | MViTac [7] | ✓ | ✓ | 60.3 |
| Supervised Learning | STAM [8] | - | ✓ | 80.0 |
| | Calandra *et. al* [12] | ✓ | ✓ | 73.1 |
| | VTFSA [9] | ✓ | ✓ | 78.1 |
| | ConViTac (**Ours**) | ✓ | ✓ | **84.3** |

where $e_i^p$ represents the $i$-th projected feature in the batch, and $\tau$ represents the temperature parameter that scales the logits. We set the diagonal elements of $S$ to $-\infty$ to prevent self-similarities. Notably, we utilize DINO [10] as $\mathcal{E}^c$.

Consequently, we freeze this pretrained $\mathcal{E}^c$ during the following training, and utilize it to project both visual and tactile data into a shared feature space to get unified latent representations. We take the projected contrastive embeddings $e^p$ as a condition to control the coupling in feature fusion through cross-modal attention [11].

## III. EXPERIMENTS

We evaluate our approach on the following tasks: 1) material property identification, specifically i) material classification, ii) discrimination between hard versus soft surfaces, and iii) distinction between smooth versus textured surfaces, and 2) robot grasping success prediction. We utilize the Touch and Go [5], the Feeling of Success [12], and the ObjectFolder Real [13] datasets in our experiments. We conduct our experiments on an NVIDIA RTX 3080Ti GPU, with a batch size of 16, using the Adam [14] optimizer for model training. The initial learning rate was set at 0.1, with the models typically converging within 30 epochs for each task and dataset, and the number of patches $P$ was fixed at 16. For baseline implementations, we followed the original papers' specifications. Model performance was evaluated using accuracy (Acc).

Our ConViTac demonstrates superior performance across multiple datasets, outperforming both contrastive and supervised learning baselines by significant margins (from 11.4%

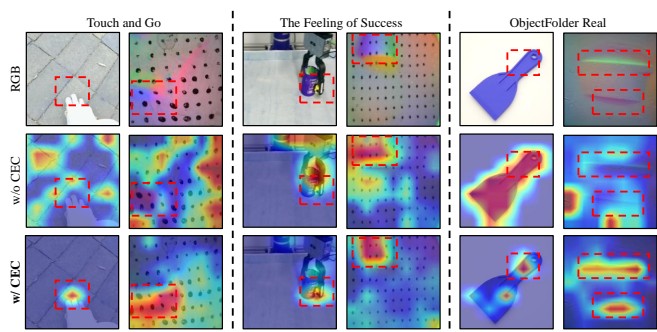

Fig. 2. Visualization for the effectiveness of our CEC mechanism using Grad-Cam [4] with highlights in red boxes. With CEC mechanism, our ConViTac pays more attention to potential contacting areas for both visual and tactile modalities, promoting their connectivity and thereby enhancing the effectiveness of visual-tactile feature fusion. In visual data, ConViTac identifies touching areas in the Touch and Go dataset [5], grasp points in the Feeling of Success dataset [12], and potential contact areas corresponding to tactile data within objects in the ObjectFolder dataset [13]. For tactile data, ConViTac with CEC mechanism also focuses more precisely on contact areas.

to 33.7% on Touch and Go and from 12.0% to 30.4% on ObjectFolder Real). This exceptional performance can be attributed to our unique approach of projecting visual-tactile inputs into latent embeddings as conditioned signals, combined with a hybrid learning strategy. Unlike traditional methods that rely solely on either contrastive or supervised learning, ConViTac leverages both paradigms by utilizing a pretrained contrastive projection encoder for feature-level optimization while maintaining cross-entropy loss for direct prediction error measurement.

## IV. CONCLUSION

In this paper, we introduce ConViTac, a novel visual-tactile representation learning framework designed to align visual-tactile fusion with contrastive representations. To be specific, we present the Contrastive Embedding Conditioning (CEC) mechanism that leverages a contrastive encoder pretrained through self-supervised contrastive learning to project visual and tactile inputs into unified latent embeddings. These embeddings are used to couple visual-tactile feature fusion through cross-modal attention, aiming at aligning the unified representations and enhancing performance on downstream tasks. We conducted extensive experiments on material identification and grasping prediction datasets, demonstrating the superiority of ConViTac over SoTA baselines.

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
