# OpenReview forum: "ConViTac: Aligning Visual-Tactile Fusion with Contrastive Representations"
_IEEE.org/IROS/2025/Workshop/Tactile_Sensing — IROS 2025 Workshop Tactile Sensing OralPoster_

### Official Review · Reviewer_MDQw · 2025-09-20
**Aligning Visual-Tactile Fusion**

**Rating:** 9
**Confidence:** 4

**Review:**

This ConViTac research shines in visual-tactile representation learning. Its Contrastive Embedding Conditioning (CEC) mechanism, using self-supervised pre-trained encoders for unified embeddings, smartly solves feature alignment issues.

---

### Official Review · Reviewer_NSz8 · 2025-09-21
**Nice paper**

**Rating:** 7
**Confidence:** 4

**Review:**

This paper presents a contrastive learning approach to learning visual-tactile representation for material and grasp success prediction. The proposed method can outperform various supervised learning and contrastive learning baselines across datasets.

Suggestions: The clarity of figure 2 can be improved (e.g. are vision and tactile branches share the same DINOv2? What are Fc / Ft? What are the dashed lines in "Cross-modal Fusion Alignment"?)